# Dual Transcriptome Analysis Reveals That *ChATG8* Is Required for Fungal Development, Melanization and Pathogenicity during the Interaction between *Colletotrichum higginsianum* and *Arabidopsis thaliana*

**DOI:** 10.3390/ijms24054376

**Published:** 2023-02-22

**Authors:** Yiming Zhu, Lingtao Duan, Chengqi Zhu, Li Wang, Zhenrui He, Mei Yang, Erxun Zhou

**Affiliations:** 1Guangdong Province Key Laboratory of Microbial Signals and Disease Control, College of Plant Protection, South China Agricultural University, Guangzhou 510642, China; 2College of Life Sciences, Zhejiang University, Hangzhou 310030, China

**Keywords:** pathogen–host interaction, *Colletotrichum higginsianum*, *ChATG8*, *ChTHR1*, dual RNA-seq

## Abstract

Anthracnose disease of cruciferous plants caused by *Colletotrichum higginsianum* is a serious fungal disease that affects cruciferous crops such as Chinese cabbage, Chinese flowering cabbage, broccoli, mustard plant, as well as the model plant *Arabidopsis thaliana*. Dual transcriptome analysis is commonly used to identify the potential mechanisms of interaction between host and pathogen. In order to identify differentially expressed genes (DEGs) in both the pathogen and host, the conidia of wild-type (ChWT) and *Chatg8* mutant (*Chatg8*Δ) strains were inoculated onto leaves of *A. thaliana*, and the infected leaves of *A. thaliana* at 8, 22, 40, and 60 h post-inoculation (hpi) were subjected to dual RNA-seq analysis. The results showed that comparison of gene expression between the ‘ChWT’ and ‘*Chatg8*Δ’ samples detected 900 DEGs (306 upregulated and 594 down-regulated) at 8 hpi, 692 DEGs (283 upregulated and 409 down-regulated) at 22 hpi, 496 DEGs (220 upregulated and 276 down-regulated) at 40 hpi, and 3159 DEGs (1544 upregulated and 1615 down-regulated) at 60 hpi. GO and KEGG analyses found that the DEGs were mainly involved in fungal development, biosynthesis of secondary metabolites, plant–fungal interactions, and phytohormone signaling. The regulatory network of key genes annotated in the Pathogen–Host Interactions database (PHI-base) and Plant Resistance Genes database (PRGdb), as well as a number of key genes highly correlated with the 8, 22, 40, and 60 hpi, were identified during the infection. Among the key genes, the most significant enrichment was in the gene encoding the trihydroxynaphthalene reductase (THR1) in the melanin biosynthesis pathway. Both *Chatg8*Δ and *Chthr1*Δ strains showed varying degrees of reduction of melanin in appressoria and colonies. The pathogenicity of the *Chthr1*Δ strain was lost. In addition, six DEGs from *C. higginsianum* and six DEGs from *A. thaliana* were selected for real-time quantitative PCR (RT-qPCR) to confirm the RNA-seq results. The information gathered from this study enriches the resources available for research into the role of the gene *ChATG8* during the infection of *A. thaliana* by *C. higginsianum*, such as potential links between melanin biosynthesis and autophagy, and the response of *A. thaliana* to different fungal strains, thereby providing a theoretical basis for the breeding of cruciferous green leaf vegetable cultivars with resistance to anthracnose disease.

## 1. Introduction

Cruciferous plants include a wide number of species that are economically and nutritionally important all over the world. This category contains vegetables such as Chinese cabbage, Chinese flowering cabbage, cabbages, broccoli, radish, cauliflower, mustards, and also oilseeds such as rapeseed and canola [1]. When cultivated on a large scale, cruciferous horticulture crops are particularly vulnerable to pathogen invasion during production. Cruciferous anthracnose disease, caused by the filamentous fungus *Colletotrichum higginsianum*, is one of the most important threats to cruciferous crops. Moreover, most *Arabidopsis* ecotypes are also susceptible to *C. higginsianum*, so *Colletotrichum–Arabidopsis* interactions can serve as a model pathological system to help us gaining more insight into the interactions between pathogens and plants [2,3]. For *C. higginsianum*, the appressoria produced by the conidia are essential structures for the infection of the host plant. Conidia began to germinate when they land on the leaves of plant and form a mature appressorium, which penetrate plant epidermal cells by a penetration peg. Following this, bulbous hyphae are formed from the penetration peg within the initially infected epidermal cells. At this stage, which is called the “biotrophic infection phase”, infected cells remain normally plasmolysed, and the host plasmalemma and tonoplast remained functional. Then, the bulbous hyphae develop rapidly to produce narrow hyphae and neighboring cells are colonized. At this stage, narrow hyphae grow rapidly and eventually cause necrotic lesions that appear as water-soaked lesions on the surface of the infected host [2,3,4,5]. At high temperatures and during wet seasons, cruciferous anthracnose disease is common in fields [4]. Currently, the use of fungicides is the primary method to control cruciferous anthracnose. Continued exposure to fungicides, on the other hand, increases the risk of environmental contamination and pathogen resistance. Gaining a better comprehension of the defense mechanisms used by cruciferous plants in response to *C. higginsianum* infection, as well as the pathogenic mechanisms of *C. higginsianum*, will allow us to breed resistant cultivars of cruciferous crops, develop novel fungicides, and design new and safer control strategies for anthracnose diseases of cruciferous crops.

Autophagy (ATG) is a conserved cytoprotective mechanism that facilitates the degradation of damaged or unwanted cellular components, which are referred to collectively as cargo. Atg8 protein plays key functions during macroautophagy, upon conjugation to double-membrane vesicles, termed autophagosomes. Finally, the autophagosomes sequester cargo for lysosomal/vacuolar degradation [6]. In many fungal pathogens, autophagy is important for pathogenicity and the gene *ATG8* is also required for conidiation and pathogenicity [7]. For example, *Pyricularia oryzae* (=*Magnaporthe oryzae*) had varied degrees of loss or decrease in conidiation and pathogenicity when it lost its autophagic core gene, *MoAtg8* [8,9,10]. *CoATG8* has been shown to be involved in conidiation and pathogenicity in the genus *Colletotrichum* [11]. Recent studies have shown that ATG8s also function in single-membrane organelles in addition to their traditional roles resulting in significantly diverse degradative or secretory fates, vesicle maturation, and cargo identification. ATG8s are associated with many vesicles through complex regulatory processes that are not being completely understood [6,12]. Therefore, a better understanding of the role of *Atg8* in the process of plant infection by pathogenic fungi is particularly important.

High-throughput sequencing (HTS) technologies allow more detailed monitoring of molecular changes in plants under various stresses. Among them, RNA-seq has been widely used in plant–pathogen interaction studies in many agricultural crops such as citrus, apple, soybean, and tomato plants [13,14,15,16]. RNA-seq was also used to study interactions between *A. thaliana* and *C. higginsianum* [3,17]. Most of the studies mentioned above were restricted to a single transcription study of *C. higginsianum*, and it is still unclear how *A. thaliana* responds to the pathogen’s attack. Recently, dual RNA-seq technology, which simultaneously sequences and analyzes the gene expression profiles of two (or more) species by simply sharing the same cDNA library, has provided us with a powerful tool to study in vivo interactions between pathogenic fungi and their host plants [18], thus revealing dynamic changes in gene expression between two interacting species [19], and specifically identify genes associated with the dynamic expression profiles of host–pathogen interactions [20]. To date, the role of *ChATG8* during the infection and counterattack response of *A. thaliana* against ChWT and *Chatg8*Δ mutant strains are unknown.

The present study used dual RNA-seq analysis to investigate the changes in *A. thaliana* infected with ChWT and *Chatg8*Δ mutant strains. cDNA libraries were constructed and further analyzed for identified DEGs. The expression of fungal genes was also investigated at four infection stages (8, 22, 40, 60 hpi) to discover potential role of the *ChATG8* gene in the infection process. Furthermore, RT-qPCR experiments were performed to validate the reliability of the dual RNA-seq data. Through this study, we hope to gain insight into the interaction between ChWT strain and *A. thaliana*, the potential pathogenic factors associated with ChAtg8 during pathogenesis, and the defense response of *A. thaliana* to help us better control cruciferous anthracnose.

## 2. Results

### 2.1. RNA-seq Statistical Analysis

When the conidia of *C. higginsianum* land on the host surface, they begin to germinate and produce germ tubes by 8 hpi. Then, the appressorium that swells from the germ tube apex fully matures at 22 hpi, followed by the biotrophic infection phase at 40 hpi, and finally the necrotrophic infection phase at 60 hpi [2,3,4]. To investigate the role of autophagy in the infection process of *C. higginsianum* and the response of *A. thaliana* to the pathogen during this process, the conidial suspensions of WT and *Chatg8*Δ mutant strains were sprayed onto *Arabidopsis* plants (Figure 1A). The results showed that there were no disease lesions on *Arabidopsis* leaves inoculated with the WT conidial suspensions at 8 hpi and 22 hpi, but water-soaked, necrotic collapsed anthracnose lesions and yellowing symptoms could be seen on the infected leaves of *Arabidopsis* plants from 40 hpi to 60 hpi. In contrast, under the same conditions, the *Chatg8*Δ mutant hardly caused any necrotic lesions to the inoculated leaves at all time points (Figure 1B). According to the observations, four time points (8, 22, 40, and 60 hpi) were chosen as sampling time points for RNA-seq analysis so as to obtain sufficient transcripts of the gene *ChATG8* in *C. higginsianum* and to investigate the dynamic transcript changes in both *A. thaliana* and the fungal strains.

Leaves infected with WT and *Chatg8*Δ strains were sampled at four time points (ChWT-8 hpi-1/2/3, ChWT-22 hpi-1/2/3, ChWT-40 hpi-1/2/3, ChWT-60 hpi-1/2/3, *Chatg8*Δ-8 hpi-1/2/3, *Chatg8*Δ-22 hpi-1/2/3, *Chatg8*Δ-40 hpi-1/2/3, *Chatg8*Δ-60 hpi-1/2/3) with three biological replicates at each time point. Appendix A displays the summary statistics of raw reads and filtered clean reads at each time point for the three replicates. The 24 samples generated 1.90 billion raw reads and 1.89 billion filtered clean reads. Furthermore, at each time point, each sample contained an average of 11.8 Gb of clean data. According to the ratios of Q20 and Q30, which were higher than 97% and 92%, respectively, the quality of the sequencing data were sufficient for further study. After redundant deletion and species identification, de novo assembly was conducted using HISAT2. 2.4 software using paired-end methods, and 23,670 unigenes of *A. thaliana* and 13,677 unigenes of *C. higginsianum* were eventually generated and utilized as reference transcripts for further research.

### 2.2. Gene Expression Changes in C. higginsianum and A. thaliana during Infection

Principal component analysis (PCA) was carried out to analyze the relationships of biological replicates in the samples as well as differences between samples. The result of PCA in *C. higginsianum* indicated that the three biological replicates at each time point clustered closely, indicating acceptable variation within the replicates at each time point (Figure 2A). Furthermore, the fungal samples could be separated into four groups. Group ‘one’ contained two samples, ChWT-8 hpi and Ch*atg8*Δ-8 hpi. This suggests that the fungal samples between WT and Ch*atg8*Δ had a similar pattern of gene expression at 8 hpi. The Group ‘one’ samples also clustered far from the others which suggest that there were distinct patterns of gene expression. Group ‘two’ contained only the sample of ChWT-60 hpi. Group ‘three’ included the closely clustered samples of Ch*atg8*Δ at 22 hpi, 40 hpi, and 60 hpi. Group ‘four’ consisted of the samples of ChWT at 22 hpi and 40 hpi (Figure 2A). In addition, hierarchical clustering (HCL) was implemented to assess the biological variability among all samples, and the results revealed a strong correlation between replicates of a single condition, and a clear separation between independent conditions (Figure 2B). Comparison of gene expression between the ‘ChWT’ and ‘*Chatg8*Δ’ sample series detected 900 DEGs (306 upregulated and 594 down-regulated) for ChWT-8 hpi vs. *Chatg8*Δ-8 hpi, 692 DEGs (283 upregulated and 409 down-regulated) for ChWT-22 hpi vs. *Chatg8*Δ-22 hpi, 496 DEGs (220 upregulated and 276 down-regulated) for ChWT-40 hpi vs. *Chatg8*Δ-40 hpi, and 3159 DEGs (1544 upregulated and 1615 down-regulated) for ChWT-60 hpi vs. *Chatg8*Δ-60 hpi (Figure 2C, Appendix A). The volcano plot shown in Figure 2D illustrates more details of the DEGs and the non-DEGs at each time point.

For *A. thaliana*, the PCA and HCL were performed on the biological variability across all samples of *A. thaliana*. The results revealed that the three biological replicates at each time point clustered closely, indicating that the variation within the replicates at each time point and condition was acceptable (Figure 2E,F). The numbers of DEGs in *A. thaliana* at each time point are shown in Figure 2G. The results showed that the numbers of DEGs increased slowly at 8, 22, and 40 hpi for ChWT vs. *Chatg8*Δ. However, the DEGs increased sharply at 60 hpi, which indicates that *A. thaliana* differed very much in transcript levels at 60 hpi with ChWT and *Chatg8*Δ (Figure 2G,H, Appendix A). Furthermore, the phenotypic differences between *A. thaliana* plants inoculated with ChWT and *Chatg8*Δ was greatest at 60 hpi, as shown in Figure 1B.

In short, the combined transcriptomic data from this series of pathogen and plant samples suggest that the transcriptional changes could support further studies. The numbers of DEGs in both the plant and pathogen samples reached the maximum at 60 hpi.

### 2.3. Functional Classifications and Pathway Analyses of the DEGs

To provide insights into the biological functions of the DEGs in the two fungal strains and *A. thaliana*, GO and KEGG enrichment analyses were performed. For the fungal strains, the GO terms of intracellular ribonucleoprotein complex, ribonucleoprotein complex, and ribosomal subunit in the cellular component terms, carboxylic acid biosynthetic process, organic acid biosynthetic process, and oxoacid metabolic process in the biological process terms, and structural molecule activity in molecular function terms were the most enriched terms in the ChWT-8 hpi vs. *Chatg8*Δ-8 hpi comparison group (Figure 3A). The 20 most enriched GO terms in the ChWT-20 hpi vs. *Chatg8*Δ-20 hpi comparison group are shown in Figure 3B. The cellular component terms of intrinsic component of membrane, membrane part, and membrane were the most enriched categories. In terms of molecular function, nucleotide binding, nucleoside phosphate binding, and transmembrane transporter activity are the most enriched categories. The single-organism process was the only biological process categories in the top 20 enriched GO terms. The 20 most significantly enriched GO terms in the ChWT-40 hpi vs. *Chatg8*Δ-40 hpi comparison group are showed in Figure 3C. The molecular functions of pattern binding, polysaccharide binding, and carbohydrate binding were the most enriched categories. The cellular component of intrinsic component of membrane, membrane part, and membrane were the most enriched categories. In terms of biological process, nucleophagy, membrane invagination, and lysosomal microautophagy were the most enriched categories. The DEGs in the ChWT-60 hpi vs. *Chatg8*Δ-60 hpi comparison group shown in Figure 3D were the most enriched in the cellular components of intracellular ribonucleoprotein complex, ribonucleoprotein complex, and preribosome, and the biological processes of purine ribonucleotide biosynthetic process, rRNA metabolic process, and purine nucleoside monophosphate metabolic process.

For *A. thaliana*, the 20 most significantly enriched GO terms at 8 hpi, 22 hpi, 40 hpi and 60 hpi were concentrated under the category of molecular functions. Figure 3E shows that the DEGs in the ChWT-8 hpi vs. *Chatg8*Δ-8 hpi comparison group were most significantly enriched in response to oxygen-containing compound, response to wounding, and response to acid chemical. The DEGs in the ChWT-22 hpi vs. *Chatg8*Δ-22 hpi comparison group were most significantly enriched in response to organic substance, response to chemical, and response to stimulus. The DEGs in the ChWT-40 hpi vs. *Chatg8*Δ-40 hpi comparison group were most significantly enriched in response to chemical, response to stimulus, and response to oxygen-containing compound. At 60 hpi, necrotic lesions were already evident on the leaves of *Arabidopsis* inoculated with the ChWT strain, but not on plants inoculated with the *Chatg8*Δ strains. The GO terms of response to stimulus, response to chemical, and oxoacid metabolic process in the biological process terms, and plastid part, chloroplast part, and plastid in the cellular component terms were the most enriched terms.

KEGG analysis assigned the DEGs in the ChWT vs. *Chatg8*Δ comparison groups from *C. higginsianum* to 371 pathways for all time points during the infection. Figure 4A shows the top 20 enriched pathways in the ChWT-8 hpi vs. *Chatg8*Δ-8 hpi comparison group. DEGs in this comparison group were most enriched in ribosome, glycerolipid metabolism, biosynthesis of secondary metabolites, 2-oxocarboxylic acid metabolism, and valine, leucine, and isoleucine biosynthesis. DEGs in the ChWT-22 hpi vs. *Chatg8*Δ-22 hpi comparison group were most enriched in ribosome, nitrogen metabolism, fatty acid degradation, alanine, aspartate, and glutamate metabolism, and glycerolipid metabolism (Figure 4B). DEGs in the ChWT-40 hpi vs. *Chatg8*Δ-40 hpi comparison group were most enriched in nitrogen metabolism, autophagy—yeast, autophagy—other eukaryotes, starch and sucrose metabolism, and methane metabolism (Figure 4C). DEGs in the ChWT-60 hpi vs. *Chatg8*Δ-60 hpi comparison group were most enriched in ribosome, ribosome biogenesis in eukaryotes, oxidative phosphorylation, one carbon pool by folate, and fructose and mannose metabolism (Figure 4D).

For *A. thaliana*, KEGG analysis assigned the DEGs in the ChWT vs. *Chatg8*Δ comparison groups to 337 pathways for all the timepoints during the infection. The DEGs were primarily enriched in phenylpropanoid biosynthesis, biosynthesis of secondary metabolites, nitrogen metabolism, starch and sucrose metabolism, and tryptophan metabolism at 8 hpi (Figure 4E); photosynthesis-antenna proteins, phenylpropanoid biosynthesis, glutathione metabolism, plant hormone signal transduction, and biosynthesis of secondary metabolites at 22 hpi (Figure 4F); zeatin biosynthesis, glutathione metabolism, plant hormone signal transduction, biosynthesis of secondary metabolites, and phenylpropanoid biosynthesis at 40 hpi (Figure 4G); and biosynthesis of secondary metabolites, metabolic pathways, photosynthesis, carbon metabolism, and biosynthesis of amino acids at 60 hpi (Figure 4H).

### 2.4. DEGs Related to Plant Defense Responses

In the phenotypic experiments above, we found that the *ChAtg8*Δ mutant did not cause necrosis in the host plant *A. thaliana*. Based on the transcriptome analysis, we hoped to answer what happens at this stage for the plant. PRGdb is a bioinformatics platform for the investigation of plant resistance genes [21]. A total of 2639 genes in *A. thaliana* that were annotated by PRGdb were analyzed for expression profiles in the order of ChWT-8 hpi, ChWT-22 hpi, ChWT-40 hpi, and ChWT-60 hpi (Appendix A). Among them, 497 genes were significantly enriched in one of the four infection stages with up-regulated expression in profiles 10, 19, 16, 6, and 13 (Figure 5E). As illustrated in Figure 5F, 135 of the 497 key genes were significantly down-regulated in the ChWT-22 hpi vs. *Chatg8*Δ-22 hpi or ChWT-40 hpi vs. *Chatg8*Δ-40 hpi comparison groups. Next, these 135 genes were investigated by GO and KEGG enrichment analyses. In this gene set, almost all of for the genes were associated with biological processes related to plant immune-related and stress responses such as defense responses, response to stimulus, response to stress, response to chemical, and immune system processes (Figure 5G). The KEGG enrichment results showed that the 135 genes were found to be mainly associated with metabolism and signaling related pathways, such as starch and sucrose metabolism, cyanoamino acid metabolism, phenylpropanoid biosynthesis, and MAPK signaling pathway—plants (Figure 5H). These results imply that the *Arabidopsis* plants inoculated with conidia of the *Chatg8*Δ strains did not seem to have an immune response.

### 2.5. DEGs Involved in Phytohormone Signaling

The related hormonal pathways were schematically illustrated in Figure 6. Many phytohormones are involved and play crucial regulatory functions in plant–pathogen interactions, including abscisic acid (ABA), auxin (AUX), ethylene (ET), jasmonic acid (JA), brassionosteroid (BR), and salicylic acid (SA) [22]. The related DEGs of several hormone signaling pathways were analyzed in the infected leaves of *Arabidopsis* plants (Figure 6 and Appendix A). Several DEGs involved in auxin signaling had differential expression, e.g., *ARF, SAUR,* and auxin-responsive *GH3* homologues, which showed notably down-regulated expression. All DEGs in the SA signaling pathway were down-regulated during the whole process and two *PR-1* homologs were significantly down-regulated at 22 hpi. The only *AHP* homolog in the cytokinine signaling pathway was down-regulated. The *PP2C* homolog involved in the ABA signaling pathway was down-regulated at 8 hpi and 60 hpi. *ERF1/2* homologs known to be ET responsive were differentially expressed; one of them was down-regulated during the whole process and the others were down-regulated at 22 hpi and 60 hpi. Previous studies revealed that BR belongs to a distinct family of growth-promoting steroid hormones, which are known to be important regulators of plant immunity [23]. The *TCH4* homolog involved in BR signaling cascades was significantly down-regulated at 22 hpi. Finally, the *JAZ* homolog in the JA signaling pathway was notably down-regulated at 60 hpi.

### 2.6. DEGs Related to Virulence in C. higginsianum

PHI-base is a database that provides regulatory information on the molecules and biology of genes that have been shown to influence the outcome of host–pathogen interactions [24]. To identify the important avirulence/virulence factors of *C. higginsianum*, 3402 key genes were annotated by PHI-base and analyzed for expression profiles in the order of ChWT-8 hpi, ChWT-22 hpi, ChWT-40 hpi, and ChWT-60 hpi (Appendix A). A total of 1018 genes were significantly enriched in one of the four infection stages with up-regulated expression in profiles 16, 10, 18, 17, 13, and 19 (Figure 5A). As previously reported, virulence factors including effectors of *C. higginsianum* mainly accumulated at the biotrophic interfacial bodies and are secreted to the host cell from the biotrophic interfacial bodies [2]. The role of autophagy in the infection of *C. higginsianum* is unknown. To investigate the role of autophagy during the transition from the biotrophic infection phase to the necrotrophic infection phase, two comparison groups (ChWT-22 hpi vs. *Chatg8*Δ-22 hpi and ChWT-40 hpi vs. *Chatg8*Δ-40 hpi) were chosen. Compared to the ChWT, a total of 111 genes were significantly down-regulated, and 82 genes were significantly down-regulated in the ChWT-22 hpi vs. *Chatg8*Δ-22 hpi comparison group. Meanwhile, 71 genes were significantly down-regulated in the ChWT-40 hpi vs. *Chatg8*Δ-40 hpi comparison group, and 42 genes were significantly down-regulated in both two comparison groups (Figure 5B). To further study the biological functions of the above 111 genes, GO and KEGG enrichment analyses were performed. Peptidase activity, acting on L-amino acid peptides, pattern binding, and polysaccharide binding were the most enriched GO terms (Figure 5C). Furthermore, the involvement of a total of 20 KEGG pathways was demonstrated. Pathways with the highest DEG representation were for ‘biosynthesis of unsaturated fatty acids’, followed by ‘pentose and glucuronate interconversions’, ‘biotin metabolism’, and ‘phenylalanine metabolism’ (Figure 5D). The above findings demonstrated that during the infection, *C. higginsianum* stimulated a number of metabolic processes that produced energy and toxic metabolites to attack host cells. Additionally, the gene *ChATG8* is highly related to these processes.

### 2.7. Loss of Atg8 Gene Affects Fungal Melanin Biosynthesis

The KEGG enrichment analysis of the 111 genes selected above revealed that the gene CH63R_08913 was most significantly enriched and was classified in the biosynthesis of unsaturated fatty acids pathway. The gene CH63R_08913 was expressed only at 22 hpi, an important time point for the melanization and maturation of *C. higginsianum* appressoria [3]. By gene sequence comparison, CH63R_08913 was found to encode a homolog of trihydroxynaphthalene reductase (THR1), which is an important enzyme in the melanin biosynthesis pathway in many fungi [25,26,27]. To investigate the relationship between the gene *ChATG8* and the biosynthesis of melanin, conidia of WT and *Chatg8Δ* strains were observed, and the result showed that more than 80% conidia of WT strain germinated properly on a hydrophobic surface to form a single unbranched germ tube subtending a single appressorium (Type 1), but 42% conidia of the *Chatg8*Δ strains produced a slender, bifurcated germ tube without appressoria (Type 2), 23% conidia of *Chatg8*Δ strains produced two germs without appressoria at one end (Type 3), nearly 20% of conidia germinated and formed appressoria without melanin (Type 4), and nearly 18% of conidia did not germinate (Type 5) (Figure 7A). This result indicates that 80% of the conidia of the *Chatg8*Δ strains were unable to form an appressorium, while about 20% of the conidia could form an appressorium, but this appressorium could not accumulate melanin normally. Next, to verify whether the gene CH63R_08913 (*ChTHR1*) also plays a role as a trihydroxynaphthalene reductase in *C. higginsianum*, we knocked down the gene CH63R_08913 (*ChTHR1*) (Appendix A). As shown in Figure 7B, the appressorium of the *Chthr1*Δ strains also failed to accumulate melanin normally. Unlike the *Chatg8*Δ strains, most germination types of *Chthr1*Δ strains conidia were only type 4, with very low numbers of types 1, 2, 3, and 5. In addition, the colony morphology of the *Chatg8*Δ strains and *Chthr1*Δ strains were also altered to a greater extent. The colonies of the *Chatg8*Δ strains showed an orange color, and the colonies of *Chthr1*Δ strains were brown, but the melanin in their colonies was significantly lower than that of WT (Figure 7C). Similar to the *Chatg8*Δ strains, the *Chthr1*Δ strains also lost pathogenicity. These results suggest that the *ChATG8* is not only involved in the pathway of melanin biosynthesis, but also affects the growth and development and pathogenicity of *C. higginsianum* in other ways.

### 2.8. Confirmation of RNA-seq Data by RT-qPCR

To ascertain the reliability of the generated dual RNA-seq data, the expression of 12 DEGs were analyzed using RT-qPCR assays, of which six were derived from *C. higginsianum* (CH63R_01708, CH63R_03174, CH63R_01918, CH63R_09049, CH63R_03670, CH63R_06437) and six were from *A. thaliana* (AT3G60120.1, AT5G50760.1, AT5G59220.1, AT1G64780.1, AT5G58310.1, AT4G35090.1) (Figure 8). Comparable up- or down-regulation expression patterns were seen between the RNA-seq and the RT-qPCR results. The RT-qPCR results indicated that these DEGs were in good agreement with the RNA-Seq results, showing a correlation coefficient (R^2^) > 0.8957 (Appendix A). Minor variations in expression levels can point to different sensitivity between the two approaches. Overall, the outcomes validated the accuracy of the RNA-seq data.

## 3. Discussion

To identify any dynamic changes in the plant tissue and gain a better understanding of the function of *ChATG8* in host–pathogen interactions, dual RNA-seq of *A. thaliana* leaves infected by the ChWT and *Chatg8*Δ strains of *C. higginsianum* was performed. This study compared the gene expression of *A. thaliana*, infected by ChWT and *Chatg8*Δ strains, at four infection stages [3,28]. A total of 15055 and 23956 sequenced genes were identified in *C. higginsianum* and *A. thaliana*, respectively. The numbers of DEGs found in *C. higginsianum* and *A. thaliana* were 5247 and 8879, respectively. For the pathogen, the analysis of these DEGs focused on the effect of the gene *ChATG8* on the pathogenicity of *C. higginsianum* during infection. For the host plant, the analysis of these DEGs focused on the changes in the immune response of *A. thaliana* to the infections of both ChWT and *Chatg8*Δ strains.

For genus *Colletotrichum* and *P. oryzae*, the most important infection structures are appressoria, which are required for infection [29]. Research in the past few decades showed that several signaling pathways, such as the MAPK pathway, TOR pathway, and autophagy pathway, were involved in appressorium formation and invasive growth [7,30,31,32]. Our study showed that *Tec1* located in the MAPK signal pathway induced by starvation was significantly down-regulated in the two comparison groups ChWT-8 hpi vs. *Chatg8*Δ-8 hpi and ChWT-22 hpi vs. *Chatg8Δ*-22 hpi. *Slt2* involved in the MAPK pathway is induced by cell wall stress and was significantly down-regulated at 22 hpi and 40 hpi. In addition, many DEGs at the four time points were involved in the phosphatidylinositol signaling pathway, which implies an important underlying link between the phosphatidylinositol signaling pathway and the autophagic pathway during *C. higginsianum* infection of host plants. 

PHI-base contains molecular and biological information on genes that have been proven to affect the outcome of pathogen–host interactions [24]. We also aligned all the DEGs to the PHI database, and then the DEGs obtained from the alignment were subjected to expression pattern analysis, and finally the DEGs whose expressions were up-regulated at any of the four time points were selected for GO and KEGG analyses. It was hypothesized that the relationship between autophagy and these pathogenic factors would be discovered. For instance, the gene CH63R_08913 (*THR1*) was the most prominently enriched gene in the biosynthesis of unsaturated fatty acids in the KEGG pathway analysis. We proved that the gene CH63R_08913 (*THR1*), encoding a homolog of trihydroxynaphthalene reductase (*THR1*), was involved in the melanin biosynthesis pathway in *C. higginsianum*. By knocking out the gene *ChTHR1*, we found that the appressoria of *Chthr1*Δ were unable to accumulate melanin, which is also one of the phenotypes of the appressoria of *Chatg8*Δ. The accumulation of melanin in *Chthr1*Δ colonies was also lower than that of WT colonies. However, their colony colors were also different from those of *Chatg8*Δ; this may be due to the fact that the gene *ChATG8* not only affected the melanin synthesis pathway, but also affected other color-related secondary metabolite pathways. In *C. lagenarium*, the mutant strain that lost the gene *THR1* formed nonmelanized appressoria, and the mutant was unable to infect its host plant because the abnormal appressoria could not penetrate the host plant’s epidermal cells [33]. This is similar to the phenotype of the *A. thaliana* that were inoculated with *Chatg8*Δ and *Chthr1*Δ in our study. In melanocytes, the autophagy proteins Atg8 and Atg4B have been reported to mediate melanosome trafficking on the cytoskeletal tracks [34]. However, to date, there are no reports related to the involvement of autophagy or ATG8 in melanin biosynthesis in fungi.

From the perspective of plants, phytohormones are critical in the developmental processes and signaling networks that regulate plant responses to various stresses [34]. Cell death and immune responses are regulated by ET signaling elements such *EIN2*, *EIN3*, *EBF1/2*, and *ERF1/2* [35]. In response to diverse biotic and abiotic challenges, JA signaling is systemically activated, boosting the resistance of host plants to some pathogens [36]. Inducing defense and resistance in response to pathogen assaults is another critical function of SA [37]. In our study, 21 DEGs involved in the auxin, SA, cytokinine, ABA, ET, BR, and JA pathways were significantly down-regulated. Their interplay induced defense responses to *C. higginsianum* infection. The involvement and characteristics of DEGs in the intricate phytohormone signaling pathways suggest that these signals were not just simple linear and isolated cascades in response to *C. higginsianum* infection, but also collaborated with one another.

To further determine the immune response of *A. thaliana* during different infection stages (8, 22, 40, and 60 hpi), all DEGs from *A. thaliana* were aligned to the phi-database, which is a bioinformatics platform for plant resistance gene analysis [21]. For example, the gene SBT3.3 (AT1G32960) of *A. thaliana*, encoding a serine protease homologous to the tomato P69C subtilase, was significantly up-regulated at 22 and 40 hpi. Previous studies found that the protein SBT3.3 may be involved in the process of pathogen recognition and activation of signaling pathways [38]. Two amidohydrolases IAR3 (AT1G51760) and ILL6 (AT1G51760), which play essential roles for proper JA–Ile homeostasis upon fungal attack, were also identified [39]. Moreover, the 135 key genes obtained by expression pattern analysis and PHI database alignment may be highly correlated with the response of *Arabidopsis* to pathogenic fungal infection.

Currently, the role of ChAtg8 in the infection of *A. thaliana* by *C. higginsianum* is unclear. Previous studies showed that loss of the gene *ATG8* results in the inability of the pathogenic fungus to successfully penetrate plant epidermal cells [29], and our study also indicated the same result. For *C. higginsianum*, by screening secondary metabolite genes and candidate pathogenesis-related genes, we confirmed the involvement of the gene *ChATG8* and autophagy in melanin biosynthesis, and provided a reference for subsequent studies. For *A. thaliana*, we screened for key genes that may be relevant to its disease resistance and contribute to our deeper understanding of the pathogen–plant interaction mechanism. In this study, a powerful methodology for dual transcriptome analysis of host plants and pathogenic fungi was designed to establish the basis for a comprehensive study of the pathogenicity-related genes and pathogenesis of cruciferous anthracnose.

## 4. Methods

### 4.1. Plant Materials and Artificial Inoculation

In all pathogenicity assays, the ecotype Col-0 of *A. thaliana* was proposed as the susceptible line. *Arabidopsis* plants were cultivated in growth chambers and four-week-old seedlings were used for inoculation assays. The genome-sequenced *C. higginsianum* strain IMI349063 [3] was kindly contributed by Prof. Junbin Huang from Huazhong Agricultural University (Wuhan, China). *C. higginsianum* strains were incubated on potato dextrose agar (PDA, potato 200 g/L, dextrose 20 g/L, agar 20 g/L) medium at 27 °C.

For artificial inoculation, strains were first incubated on PDA in dark conditions in a 27 °C incubator for 7 d before conidia were harvested from the fungal colony of PDA plates and suspended in 5 mL of sterile distilled water. Conidial suspensions were checked with a hemocytometer and diluted to a final concentration of 1 × 10^5^ conidia/mL with sterile distilled water. Subsequently, the leaves of *A. thaliana* plants in a pot were uniformly sprayed with 10 mL of the conidial suspension. The inoculated plants were placed in a 12 h/12 h light/dark dew chamber at 27 °C with almost 100% relative humidity and all samples of inoculated leaves were harvested at 8, 22, 40, and 60 hpi (three independent biological replicates for the two treatments of leaves infected by ChWT and *Chatg8*Δ strains), and immediately frozen in liquid nitrogen and stored at −80 °C until further use.

### 4.2. Nucleic Acid Manipulation, Southern Blot

Fungal genomic DNA was extracted using the Ezup Column Fungi Genomic DNA Purification Kit (Sangon Biotech, Shanghai, China). PCR amplification was performed using Phanta Max Super-Fidelity DNA Polymerase (Vazyme Biotech, Nanjing, China). The Universal UNlQ-10 Column DNA Purification Kit (Sangon Biotech, Shanghai, China) was used to purify the PCR product.

In the Southern blot assay, NEB (MA, USA) restriction enzymes *Bsu36*I and *EcoR*I were used for digestion of genomic DNA. The DIG-High Prime DNA Labeling and Detection Starter Kit I (Roche, IN, USA) was used for probe labeling. Amersham Hybond TM-N (GE Healthcare, WI, USA) membrane was used for blotting. NBT/BCIP Stock Solution (Roche, IN, USA) was used as the probe for protein detection.

### 4.3. Deletion and Complementation of ChTHR1

To replacement of *ChTHR1* gene by the *HPH1* (hygromycin phosphotransferase) gene follows the strategy described in [40,41] (Appendix A). Fragments approximately 1500 bp in size of the upstream and downstream sequences flanking the gene were amplified with primers THR1upF/THR1upR and THR1dsF/THR1dsR (Appendix A). The Agrobacterium Transfer-DNA vector pFGL821 was digested with *Hind*III (NEB, MA, USA) and *EcoR*I (NEB, MA, USA), and then the upstream and downstream PCR products were fused into this vector to constitute the *ChTHR1* knockout vector, named p821-*ChTHR1*KO.

The construct was transferred into *Agrobacterium tumefaciens* strains AGL1, then transformed into the WT strains using *Agrobacterium tumefaciens*-mediated transformation (ATMT). After transforming the replacement vector into WT strains, hygromycin-resistant transformants were isolated and tested for resistance to hygromycin and confirmed by Southern blotting (Appendix A) [42].

### 4.4. RNA-seq and Data Assembly

Total RNA extraction and quality assessment, cDNA libraries preparation, data assembly, sequence alignment of reference genomes, and unigene annotation were performed by Gene Denovo Biotechnology Co. (Guangzhou, China). Following the manufacturer’s instructions, total RNA was extracted from each sample using Trizol reagent (Invitrogen, MA, USA). The cDNA libraries were sequenced using an Illumina NovaSeq™ 4000 instrument that generated paired-end reads lengths of 200 bp, and the clean reads were aligned to the *C. higginsianum* or *A. thaliana* genome assembly with HISAT2 [43]. Stringtie was used to reconstruct the transcripts and calculate the expression of all genes in each sample, presented as FPKM value (Appendix A) [44].

### 4.5. Differentially Expressed Genes (DEGs) and Gene Annotation Analysis

DESeq2 [45] was used to identify DEGs with an FDR (false discovery rate) < 0.05 and |log2FC| > 1. Then, DEGs were analyzed for enrichment of Gene Ontology (GO, http://www.geneontology.org/, accessed on 15 Octorber 2022) terms and Kyoto Encyclopedia of Genes and Genomes (KEGG, https://www.kegg.jp/, accessed on 15 Octorber 2022) pathways.

### 4.6. Key Genes Annotated in PHI and PRG Databases

Key genes from *C. higginsianum* were annotated by the PHI-base (Pathogen–Host Interactions database, www.phibase.org/, accessed on 2 September 2022) (Appendix A), which incorporates molecular and biological information on genes proven to influence the outcome of pathogen–host interactions [24]. Key genes from *A. thaliana* were annotated by the PRGdb (Plant Resistance Genes database, www.prgdb.org/prgdb4/, accessed on 2 September 2022) (Appendix A), which is a bioinformatics platform for the analysis of plant resistance genes [21].

### 4.7. Gene Expression Validation

The RNA-seq data were confirmed by selecting 12 DEGs and measuring their expression levels by RT-qPCR. Following the manufacturer’s instructions, cDNAs were synthesized from total RNA (1 μg) using a *TransScript*^®^ One-Step gDNA Removal and cDNA Synthesis SuperMix (TransGen Biotech, Beijing, China) in a reaction containing Anchored Oligo(dT)_18_ Primer, 2 × TS Reaction Mix, *TransScript*^®^ RT/RI Enzyme Mix, gDNA Remover, RNase-free Water, and total RNA. The ChamQ Universal SYBR qPCR Master Mix (Vazyme, Nanjing, China) was used to perform RT-qPCR on a Bio-Rad CFX96 Real-Time PCR System according to the manufacturer’s instructions. The PCR reactions were performed in a total volume of 20 μL including 10 μL 2 × ChamQ Universal SYBR qPCR Master Mix, 0.4 μL Primer-F (10 µM), 0.4 μL Primer-R (10 µM), 6 μL cDNA (50 ng/μL), and 3.2 μL ddH_2_O. The RT-qPCR program included an initial denaturation step at 95 °C for 30 s, followed by 40 cycles of 10 s at 95 °C and 30 s at 60 °C. The expression levels were normalized to the expression of the reference genes *ChACT* (CH63R_04240) [42] and *TUB2* (NM_125664.4) [46]. The relative expression levels of the genes were calculated with the formula 2^−ΔΔCt^. The expression levels of each gene were expressed as a ratio relative to the stages of infection (ChWT-8 hpi), which was set as 1. All primers used in this study are listed in Appendix A.

## 5. Conclusions

In this study, the leaf samples of *A. thaliana* inoculated with ChWT and *Chatg8*Δ strains at four infection stages, i.e., 8 hpi (germ tube emergence), 22 hpi (appressorial matured), 40 hpi (biotrophic infection phase), and 60 hpi (necrotrophic infection phase), were used for dual RNA-seq analysis. A total of 900, 692, 496, and 3149 DEGs of *C. higginsianum* and 285, 575, 971, and 7048 DEGs of *A. thaliana* were identified in the ChWT vs. *Chatg8*Δ-8, 22, 40, 60 hpi comparison groups, respectively. During *Arabidopsis* infection, a series of key genes highly correlated at the 8, 22, 40, or 60 hpi timepoints and annotated in PRGdb or PHI-base were identified. Highly correlated genes were identified at 60 hpi, expanding our understanding of the role of the autophagy-related gene *ChATG8* in *C. higginsianum* and the changes in *A. thaliana* inoculated with ChWT or *Chatg8*Δ mutants. We found that ChAtg8 affects ChThr1 and is involved in melanin biosynthesis. Additionally, six DEGs each from *C. higginsianum* and *A. thaliana* were selected for RT-qPCR assays to validate the output of the RNA-seq. In summary, this work enriches the resources available for research into the role of *ChATG8* during *A. thaliana* infection and the response of *A. thaliana* to different fungal strains, thereby providing a theoretical basis for the breeding of resistant cultivars of cruciferous crops, and to develop novel fungicides as well as to design new and safer control strategies for anthracnose diseases of cruciferous crops.

## Figures and Tables

**Figure 1 ijms-24-04376-f001:**
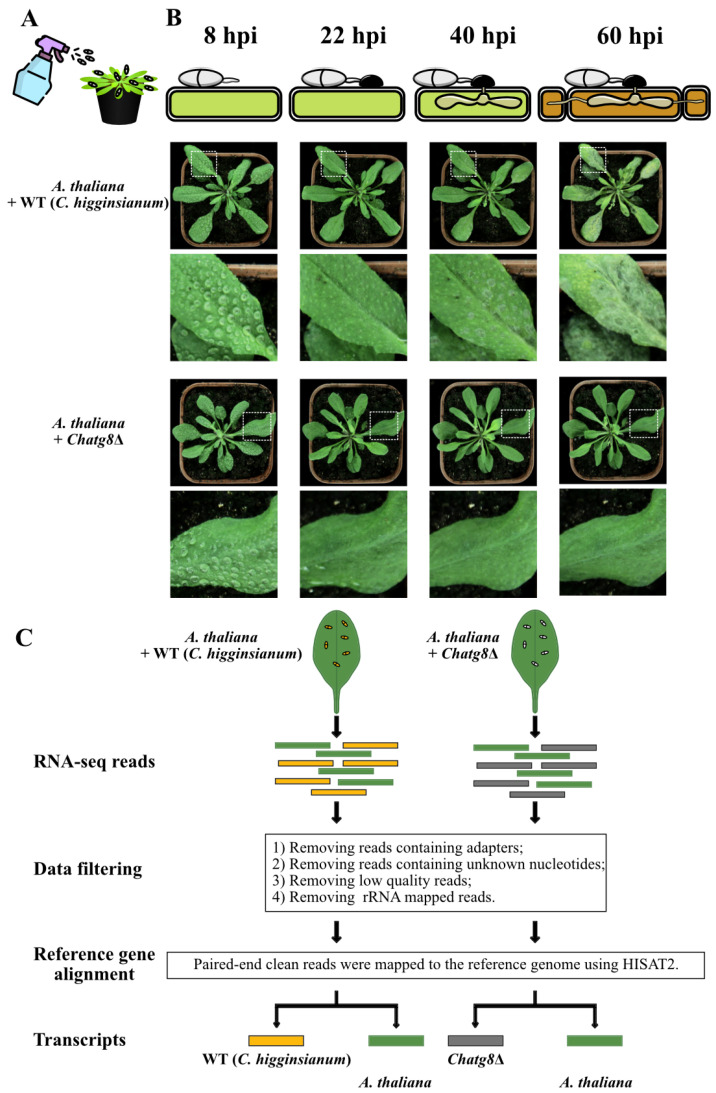
Schematic overview of dual RNA-seq analysis process. (**A**) Schematic diagram of inoculation. (**B**) Schematic illustration of the four developmental stages of *C. higginsianum* and the appearance of *A. thaliana* upon spray-inoculation with ChWT and *Chatg8*Δ conidial suspensions. In the top part of schematic diagram, the green rectangles indicate living plant cells and brown ones indicate dead plant cells. hpi is hours post-inoculation. The middle image shows *A. thaliana* upon spray-inoculation with ChWT strain at 8, 22, 40, and 60 hpi. The bottom image shows *A. thaliana* upon spray-inoculation with *Chatg8Δ* strain at 8, 22, 40, and 60 hpi. The white dashed box indicates the enlarged leaf area below. (**C**) Flow chart represents the dual RNA-seq analysis of the mixed transcriptome obtained from *A. thaliana* leaves infected with ChWT strains and *A. thaliana* leaves infected with *Chatg8*Δ strains.

**Figure 2 ijms-24-04376-f002:**
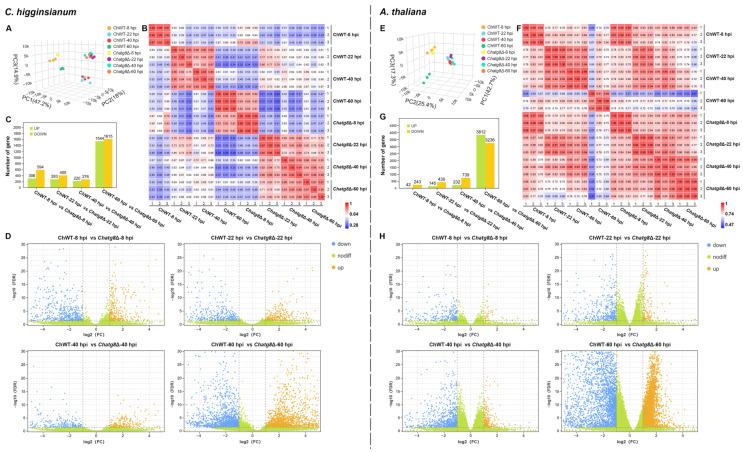
Overview of the transcriptome sequencing from *C. higginsianum* and *A. thaliana* at the four infection stages after inoculation with ChWT and *Chatg8*Δ strains. (**A**,**E**) PCA analysis among the twenty-four samples from *C. higginsianum* and *A. thaliana*. (**B**,**F**) Pearson correlation between sample analysis from *C. higginsianum* and *A. thaliana*. (**C**,**G**) Number of up- and down-regulated genes in ChWT-8 hpi vs. *Chatg8*Δ-8 hpi, ChWT-22 hpi vs. *Chatg8*Δ-22 hpi, ChWT-40 hpi vs. *Chatg8*Δ-40 hpi, and ChWT-60 hpi vs. *Chatg8*Δ-60 hpi in *C. higginsianum* and *A. thaliana*. (**D**,**H**) Volcano plot of DEGs in the four comparison groups in *C. higginsianum* and *A. thaliana*. The horizontal axes are log2 (Fold Change) and the vertical axes are −log10 (FDR). The two vertical dashed lines are the threshold for 2-fold differences, and the horizontal dashed lines are FDR value 0.05 thresholds. An orange dot indicates an upregulated gene, a blue dot indicates a downregulated gene, and a green dot represents a gene with a non-significant difference in expression.

**Figure 3 ijms-24-04376-f003:**
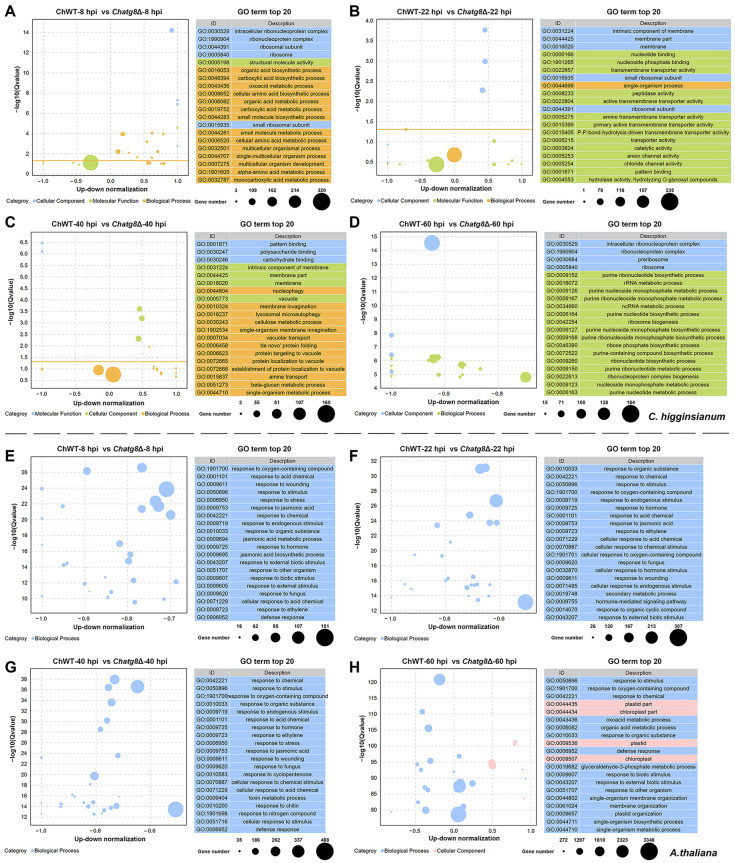
GO terms of the DEGs in the four comparison groups. (**A**,**E**) ChWT-8 hpi vs. *Chatg8*Δ-8 hpi in *C. higginsianum* or *A. thaliana*. (**B**,**F**) ChWT-22 hpi vs. *Chatg8*Δ-22 hpi in *C. higginsianum* or *A. thaliana*. (**C**,**G**) ChWT-40 hpi vs. *Chatg8*Δ-40 hpi in *C. higginsianum* or *A. thaliana*. (**D**,**H**) ChWT-60 hpi vs. *Chatg8*Δ-60 hpi in *C. higginsianum* or *A. thaliana*. The horizontal axes are up-down normalization and the vertical axes are −log10 (Qvalue). Up-down normalization refers to the proportion of the difference between the number of up-regulated genes and the number of down-regulated genes in the total number of differential genes, and the size of the dots refers to the number of DEGs in each GO term.

**Figure 4 ijms-24-04376-f004:**
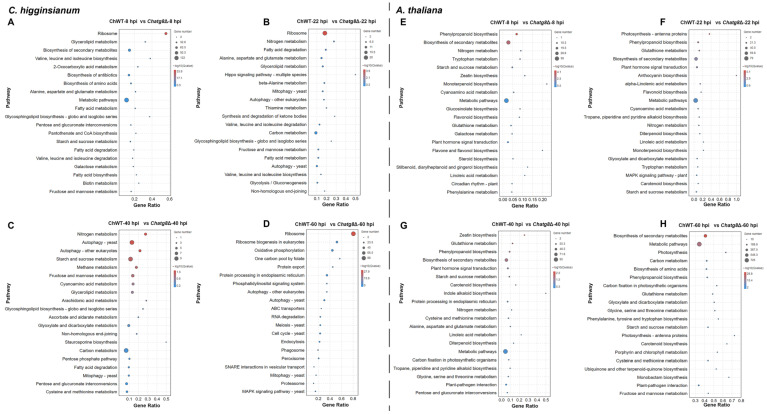
KEGG pathway classification of the DEGs in the four comparison groups. (**A**,**E**) ChWT-8 hpi vs. *Chatg8*Δ-8 hpi in *C. higginsianum* or *A. thaliana*. (**B**,**F**) ChWT-22 hpi vs. *Chatg8*Δ-22 hpi in *C. higginsianum* or *A. thaliana*. (**C**,**G**) ChWT-40 hpi vs. *Chatg8*Δ-40 hpi in *C. higginsianum* or *A. thaliana*. (**D**,**H**) ChWT-60 hpi vs. *Chatg8*Δ-60 hpi in *C. higginsianum* or *A. thaliana*. Gene ratio refers to the ratio of the number of genes associated with a KEGG pathway to the total number of DEGs and the size of the dots refers to the number of DEGs in each KEGG pathway.

**Figure 5 ijms-24-04376-f005:**
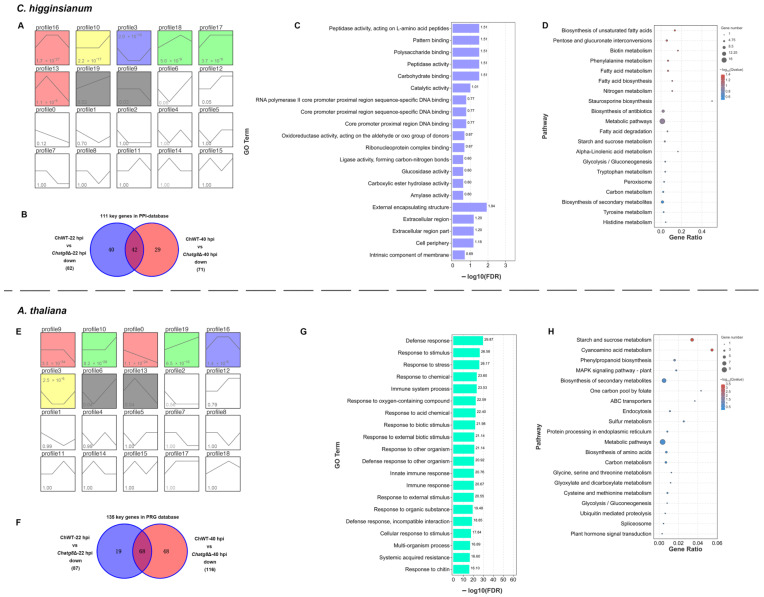
Key genes from *C. higginsianum* were annotated via PHI-base and key genes from *A. thaliana* were annotated via PRGdb. In *C. higginsianum*: (**A**) the 3402 key genes in *C. higginsianum* annotated by PHI-base were analyzed for expression profiles in the order of ChWT-8 hpi, ChWT-22 hpi, ChWT-40 hpi, and ChWT-60 hpi. A total of 1018 genes were significantly enriched in one of the four infection stages with up-regulated expression in profiles 16, 10, 18, 17, 13, and 19. (**B**) Venn diagram showing that 111 of the 1018 key genes were significantly down-regulated in the ChWT-22 hpi vs. *Chatg8*Δ-22 hpi or ChWT-40 hpi vs. *Chatg8*Δ-40 hpi comparison groups. (**C**) GO function and (**D**) KEGG pathway classification of the 111 DEGs. In *A. thaliana*: (**E**) the 2639 key genes in *A. thaliana* annotated by PRGdb were analyzed for expression profiles in the order of ChWT-8 hpi, ChWT-22 hpi, ChWT-40 hpi, and ChWT-60 hpi. A total of 497 genes were significantly enriched in one of the four infection stages with up-regulated expression in profiles 10, 19, 16, 6, and 13. (**F**) Venn diagram showing that 135 of the 497 key genes were significantly down-regulated in the ChWT-22 hpi vs. *Chatg8*Δ-22 hpi or ChWT-40 hpi vs. *Chatg8*Δ-40 hpi comparison groups. (**G**) GO functions and (**H**) KEGG pathway classification of the 135 DEGs. Gene ratio refers to the number of genes associated with a KEGG pathway to the number of DEGs and the size of the dots refers to the number of DEGs in each KEGG pathway.

**Figure 6 ijms-24-04376-f006:**
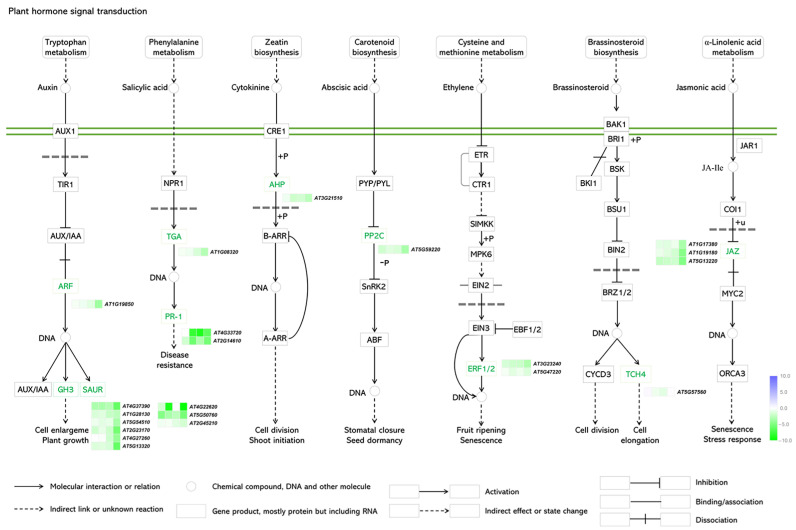
DEGs involved in plant hormone signal transduction pathways. From left to right: auxin (AUX), salicylic acid (SA), cytokinine (CTK), abscisic acid (ABA), ethylene (ETH), brassinosteroid (BR), and jasmonic acid (JA). In heatmap, each column represents log_2_ (from left to right: ChWT-8 hpi vs. *Chatg8*Δ-8 hpi, ChWT-22 hpi vs. *Chatg8*Δ-22 hpi, ChWT-40 hpi vs. *Chatg8*Δ-40 hpi, and ChWT-60 hpi vs. *Chatg8*Δ-60 hpi) and each row represents a gene. Expression values are presented as log2 fold change (purple represents up-regulation; green represents down-regulation.

**Figure 7 ijms-24-04376-f007:**
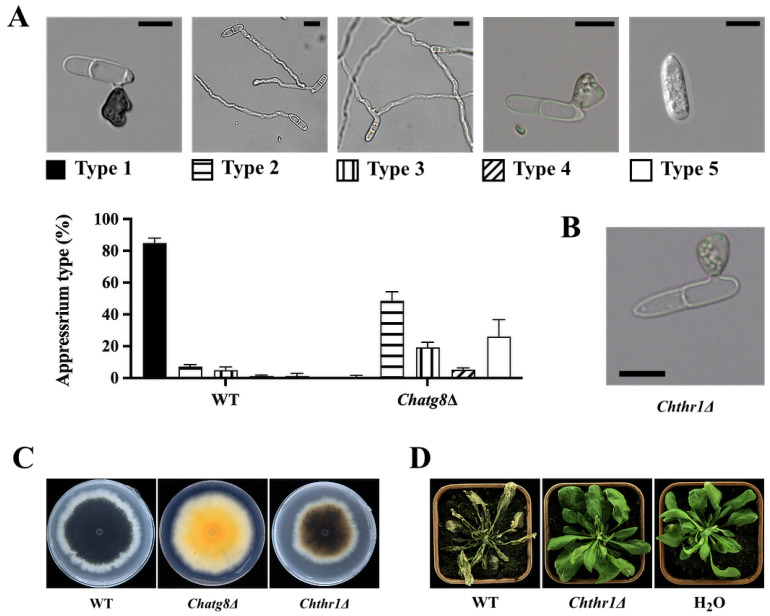
Loss of *ChATG8* or *ChTHR1* gene resulted in reduced appressoria and colony melanin and loss of pathogenicity. (**A**) Micrographs show different types of appressorial development in WT and *Chatg8*Δ strains on hydrophobic coverslips at 24 hpi at 27 °C in distilled water. The percentage of different types of appressorial development at 24 hpi was recorded from a sample of 100 conidia. The bar charts show the mean and standard deviation from three independent replications of the experiment. Black scale bar = 10 μm. (**B**) Micrographs show different types of appressorial development in *Chthr1*Δ strains on hydrophobic coverslips at 24 hpi at 27 °C in distilled water. Black scale bar = 10 μm. (**C**) Colony morphology of the WT, *Chatg8*Δ, and *Chthr1*Δ strains grown on PDA medium for 7 d. (**D**) *Arabidopsis* plants were inoculated by spraying conidial suspensions of indicated strains or sterilized water (H_2_O). Inoculated plants were photographed at 5 d post inoculation.

**Figure 8 ijms-24-04376-f008:**
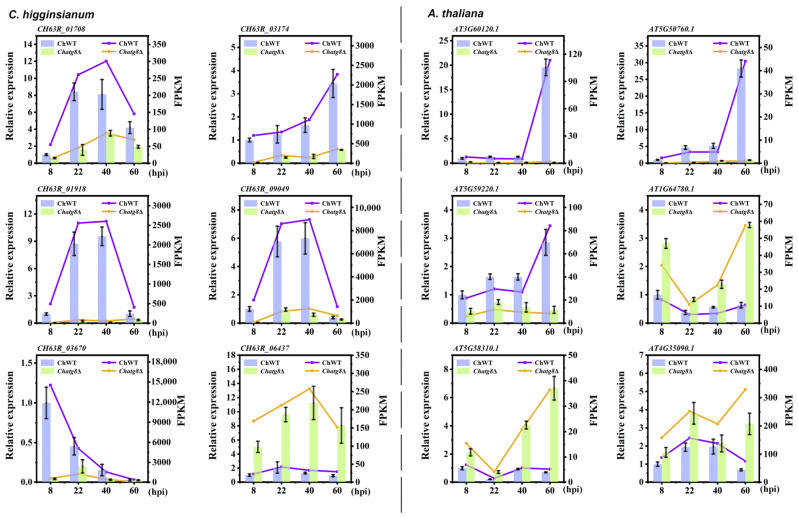
Confirmation of RNA-seq via RT-qPCR analysis. The expression of twelve selected DEGs from the RNA-seq analysis were measured by RT-qPCR in ChWT-8 hpi, *Chatg8*Δ-8 hpi, ChWT-22 hpi, *Chatg8*Δ-22 hpi, ChWT-40 hpi, *Chatg8*Δ-40 hpi, ChWT-60 hpi, and *Chatg8*Δ-60 hpi. The histograms were plotted using data obtained by RT-qPCR and the corresponded line chart was plotted by FPKM values from the RNA-seq analysis, with different colors indicating different samples. Each bar represents the mean value with standard errors of three independent experiments.

## Data Availability

The raw data of sequenced transcriptome have been deposited to the Sequence Read Archive (SRA) at NCBI under accession number PRJNA874673.

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
