# Peer review of "Dual Transcriptome Analysis Reveals That *ChATG8* Is Required for Fungal Development, Melanization and Pathogenicity during the Interaction between *Colletotrichum higginsianum* and *Arabidopsis thaliana"

_ijms, 2023, doi:10.3390/ijms24054376_

Round 1
Reviewer 1 Report (Previous Reviewer 2)
Dear authors,
You have revised most of the manuscript as suggested. However, in Table S3 there is no information on the amplicon lenght, amplification efficiency and Ta and Tm temperatures as suggested. I strongly advise you to add this info since it is crucial for results replication.
Author Response
Q: Regarding the primers in Table S3, you should add the following information: amplicon lenght, amplification efficiency, Ta and Tm temperatures.
A: Thank you for your valuable comments. We have updated the Table S3 with information on amplicon length, amplification efficiency and Ta and Tm temperatures.

Reviewer 2 Report (Previous Reviewer 3)
Authors have answered to my concerns and suggestions. Th manuscript was improved according the suggestions.
Author Response
Thank you so much for your comments/suggestions to our manuscript.
This manuscript is a resubmission of an earlier submission. The following is a list of the peer review reports and author responses from that submission.
Round 1
Reviewer 1 Report
The paper provides a detailed analysis of the transcriptomic response of both a pathogen and its host during the infection process. It is an important question although a- model organism was used and at this stage these analyses are outdated and not novel. The point of novelty was in the introduction of a mutation in autophagy of the pathogen. Although detailed and high quality analyses the results advance the field only a bit. So many modules are up and down regulated some o them are so general, some of them are really no big surprise. I could not find any clear hypothesis regarding the role of autophagy in each of the steps and definitely no testing any hypothesis.
Minor comment: the figures are not informative, DEG modules are enriched, up? down? what is the meaning of the axis, what is the meaning of the different size of the dots?
Author Response
The paper provides a detailed analysis of the transcriptomic response of both a pathogen and its host during the infection process. It is an important question although a model organism was used and at this stage these analyses are outdated and not novel. The point of novelty was in the introduction of a mutation in autophagy of the pathogen. Although detailed and high quality analyses the results advance the field only a bit. So many modules are up and down regulated some of them are so general, some of them are really no big surprise. I could not find any clear hypothesis regarding the role of autophagy in each of the steps and definitely no testing any hypothesis.
Q: Minor comment: the figures are not informative, DEG modules are enriched, up? down? what is the meaning of the axis, what is the meaning of the different size of the dots?
A: We thank the reviewer for this valuable feedback. Our research is the first to report that the loss of pathogenicity in ATG8-deficient of Colletotrichum higginsianum is likely due to the loss of melanin. ATG8 has a potential regulatory relationship with the THR1 gene, a key gene in the melanin biosynthesis pathway. This regulatory relationship has not been reported in any previous studies. The detailed regulatory mechanism between autophagy and melanin is under further investigation in authors’ laboratory.
We also thank the reviewer for constructive comments on our manuscript. We have corrected the errors that existed in Figure 3 and also updated the contents of the figure legends in Figures 2, 3, 4, 5, 6 and 7.
Reviewer 2 Report
Dear authors,
Please find below my suggestions for the improvement of your manuscript.
1. Written language should be improved, particularly the introduction section;
2. Lines 49-59: this section seems to be misplaced here; it should be in the results section since, as the time-points you present are the same of your study, this appears to be a description of what you observed;
3.Throughout the text: please check "Arabidopsis"; sometimes it is italicized and shouldn't;
4. Throughout the text: sometimes you refer to qRT-PCR but what you've done was RT-qPCR, please correct;
5. Figure 1 is very nicely constructed but you should show leaf symptoms more clearly, not just in schemes. Please add pictures of leaf symptoms;
6. Lines 137-139: This is confusing
7. Line 218: Do you mean biological process?
8. Figure 7: please add information on error bars to the legend
9. Line 398: Replace "It was hoped" by "It was hypothesized"
10. In RT-qPCR materials and methods section you should specify how much RNA you used to synthesize cDNA; components of cDNA synthesis reaction etc. Regarding the primers in Table S3, you should add the following information: amplicon lenght, amplification efficiency, Ta and Tm temperatures. You should also add statistical analysis information.
Author Response
Dear authors,
Please find below my suggestions for the improvement of your manuscript.
Q1. Written language should be improved, particularly the introduction section;
A1: Thanks for the suggestions you gave. This paper, especially the introductory section, has been carefully revised so as to further improve grammar and readability.
Q2. Lines 49-59: this section seems to be misplaced here; it should be in the results section since, as the time-points you present are the same of your study, this appears to be a description of what you observed;
A2: We are grateful to the reviewer for pointing out this error. We fully accepted the reviewer's suggestions. So, we have removed the description of the time-points (Line 53-60).
Q3.Throughout the text: please check "Arabidopsis"; sometimes it is italicized and shouldn't;
A3: Thank you so much for pointing out this error. We have changed "Arabidopsis" to italics throughout the text.
Q4. Throughout the text: sometimes you refer to qRT-PCR but what you've done was RT-qPCR, please correct;
A4: Many thanks to the reviewer for pointing out the error. We have changed "qRT-PCR" to "RT-qPCR" throughout the text.
Q5. Figure 1 is very nicely constructed but you should show leaf symptoms more clearly, not just in schemes. Please add pictures of leaf symptoms;
A5: Thank you for your valuable suggestion. As suggested by the reviewer, we have added some pictures of leaf symptoms to the corresponding position in the text (Figure 1B)
Q6. Lines 137-139: This is confusing
A6: Thank you for your comment. We have revised the phrase to make it easier to understand (Line 197-198).
Q7. Line 218: Do you mean biological process?
A7: Yes. Reactions to chemical, oxoacid metabolic processes are parts of the biological process.
Q8. Figure 7: please add information on error bars to the legend
A8: Thank you for your suggestion. We have added the information of error bars to the legend of Figure 7.
Q9. Line 398: Replace "It was hoped" by "It was hypothesized"
A9: Thanks for your suggestion. We have made the corresponding changes in line 491.
Q10. In RT-qPCR materials and methods section you should specify how much RNA you used to synthesize cDNA; components of cDNA synthesis reaction etc. Regarding the primers in Table S3, you should add the following information: amplicon lenght, amplification efficiency, Ta and Tm temperatures. You should also add statistical analysis information.
A10: Thank you for your valuable comments. We have added the details of the RT-qPCR experiment from 586-598. We have also updated the Table S3 with information on amplicon length, amplification efficiency and Ta and Tm temperatures.
Reviewer 3 Report
Authors described a dual transcriptome analysis of A. thaliana- C. higginsianum molecular interaction. The paper is well written and presented and the experimental design was appropriate. I have minor aspects that may improve the manuscript:
-Check the Latin names for genera and species throughout the manuscript, some ar not in italic form
-Increase the quality and size of some graphs: Figure 2,B and F; Figure 4; Figure 5, B and F;
-I have found many methodology details in "Results" and "Discussion" sections. I understand that some information of methodology should be provider to the reader, since M&M is presented in the end of the paper. However, I advice author to revise these sections in order to keep only the essencial methodology to understand the context of the sentences/paragraphs. All methodology details should be described in M&M
Author Response
Authors described a dual transcriptome analysis of A. thaliana- C. higginsianum molecular interaction. The paper is well written and presented and the experimental design was appropriate. I have minor aspects that may improve the manuscript:
Q1-Check the Latin names for genera and species throughout the manuscript, some ar not in italic form
A1: Thank you for your suggestion. We have changed "Arabidopsis" to italics throughout the text.
Q2-Increase the quality and size of some graphs: Figure 2,B and F; Figure 4; Figure 5, B and F;
A2: Thanks for your suggestions. We have improved the quality of graphs 2, 4 and 5 and made them clearer.
Q3-I have found many methodology details in "Results" and "Discussion" sections. I understand that some information of methodology should be provider to the reader, since M&M is presented in the end of the paper. However, I advice author to revise these sections in order to keep only the essencial methodology to understand the context of the sentences/paragraphs. All methodology details should be described in M&M
A3: Thank you so much for your valuable suggestion. We have removed the description of the experimental details from the Results and Discussion section.